# Effects of Weaning Modes on the Intestinal pH, Activity of Digestive Enzymes, and Intestinal Morphology of Piglets

**DOI:** 10.3390/ani12172200

**Published:** 2022-08-26

**Authors:** Zhifang Shi, Tianwu Wang, Jingjing Kang, Yidan Li, Yang Li, Lei Xi

**Affiliations:** 1College of Animal Science and Technology, Henan University of Animal Husbandry and Economy, Zhengzhou 450046, China; 2Faculty of Animal Science and Technology, Yunnan Agricultural University, Kunming 650210, China; 3College of Veterinary Medicine, Henan University of Animal Husbandry and Economy, Zhengzhou 450046, China; 4College of Animal Science and Technology, Jilin Agricultural University, Changchun 130118, China

**Keywords:** weaning stress, weaning mode, pH, digestive enzymes, intestinal morphology

## Abstract

**Simple Summary:**

Previous research has discovered that various weaning and mixing modes reduce weaning stress in piglets. However, there are not as many existing reports on whether relieving weaning stress with different weaning and mixing modes can promote the intestinal health of piglets. The results of this study showed that two weaning modes, 2% (*v*/*v*) glutaraldehyde spray, and weaning after mixing, can increase the daily weight gain of piglets, lower their intestinal pH, boost the activity of their digestive enzymes, and improve intestinal morphology, compared with conventional weaning modes. These discoveries indicate that these two weaning modes can reduce weaning stress to a certain extent.

**Abstract:**

The present study investigated the effects of weaning modes on the intestinal pH, digestive enzyme activities, and intestinal morphology of piglets. A total of 27 litters of “Duroc boars × Landrace × Large White” piglets were selected and randomly divided into three groups. Those groups were the conventional weaning group (C), the odorant spray group (S), and the weaning-after-mixing group (M), with three repeats in each group and three litters in each repeat. The experiment began seven days before weaning, and ended seven days after weaning. The piglets were euthanized on the eighth day after weaning, and the jejunum and ileum tissues and contents were sampled. The pH and enzyme activities of the intestinal contents were determined, and the intestinal morphologies were revealed using H&E staining. It was concluded that a 2% glutaraldehyde spray on the body surfaces of piglets after weaning could increase average weight after nursery, lower intestinal pH, increase the activities of digestive enzymes, improve intestinal morphology, and relieve weaning stress. It was also found that weaning after mixing could increase the average weight after nursery, lower intestinal pH, improve intestinal morphology, and reduce weaning stress.

## 1. Introduction

In the raising and breeding of pigs, it is very important to increase the survival rate of weaned piglets. Weaning itself is a stress, which creates higher and more precise requirements for piglet weaning. To reduce unnecessary losses due to weaning stress, appropriate weaning modes should be adopted considering the actual production situation [1]. At present, the conventional weaning modes used by farms, such as batch weaning, gradual weaning, and abrupt weaning, all cause a certain degree of weaning stress to piglets [2,3]. Since the digestive and immune systems of young piglets have not yet reached maturity, conventional weaning modes can easily lead to gastrointestinal (GI) disorders [4], reduced feed intake, indigestion, increased diarrhea, and even death, thus causing severe economic losses to pig farmers [5]. Nevertheless, piglet weaning stress could be reduced by adjusting the composition of their diets [6]. In pig farming, feed additives such as antibiotics are often used to reduce weaning stress and promote growth. However, as food safety has attracted increasing attention from both the general public and researchers, the use of antibiotic additives in pig diets has come under criticism. The addition of antibiotics to feed not only brings hidden dangers to the safety of otherwise healthy foods, but also poses potential risks to human health. Therefore, the government of China eliminated antibiotic feed additives on 1 July 2020 [7]. The Chinese government’s decision is considered a breakthrough in efforts to reduce antibiotic overuse, maintaining animal-source food safety, and ensuring public health and safety [7]. However, eliminating antibiotics in feed additives has raised new challenges with regard to reducing animal stress and implementing healthy animal farming. Therefore, ensuring the quality and safety of live pigs and developing new methods of relieving piglet weaning stress have become important areas of study for animal husbandry researchers. Ma et al. (2017) discovered that different weaning and mixing modes affect the immunity and antioxidant capacity of suckling piglets to various degrees, and thus reduce weaning stress on piglets [8]. However, it has not yet been reported whether relieving weaning stress by olfactory interference can promote the intestinal health of piglets. In this study, glutaraldehyde was used as an odorant spray. Glutaraldehyde is a broad-spectrum fungicide that can kill bacteria, viruses, spores, etc. [9]. It has the advantages of safety, broad-spectrum, high efficiency, low irritation, etc. The fungicidal effect is optimal at a concentration of 2% (*v*/*v*) and is widely used in China [9]. We hypothesized that using glutaraldehyde in different weaning modes would have different effects on the intestinal pH, digestive enzyme activity, and intestinal morphology of piglets. These differences could alleviate weaning stress in piglets and could promote piglets’ intestinal health. Therefore, the present study explores the effects of different weaning modes (odorant spray, weaning after mixing, and conventional weaning) on the pH, digestive enzyme activities, and intestinal morphologies of piglet GI tracts to find an effective weaning mode to reduce weaning stress.

## 2. Materials and Methods

### 2.1. Experimental Design

The experiment was conducted on a pig farm in Qi County, Hebi City, Henan Province. The experimental design was the same as the one published in reference [10]. In this study, a total of 27 litters of Duroc boars × Landrace × Large White ternary hybrid piglets were selected. The piglets were randomly divided into three groups: the conventional weaning group (C, control), the odorant spray group (S), and the weaning-after-mixing group (M). Three replicates were set in each group, and three litters were included in each repeat. The gender of the experimental group piglets in any given repeat was completely randomized; such a design excludes the impact of gender on results. The piglets in groups C and S were weaned separately and mixed at 28 days of age. At this stage, the latter group of piglets was sprayed with 2% (*v*/*v*) glutaraldehyde daily for seven consecutive days. The piglets in group M were mixed at the age of 21 days by removing the dividers between the three adjacent nurseries so that the piglets were mixed in situ. Then, the piglets were weaned at the age of 28 days. The experiment began seven days before weaning and ended seven days after weaning, for a total of 14 days. All experimental protocols used in this experiment were in accordance with those approved by the Henan University of Animal Husbandry and Economics’ Institutional Animal Care and Use Committee (protocol number HNUAHE 479), and institutional safety procedures were followed.

### 2.2. Experimental Diets and Rearing Conditions

During the experiment, the piglets’ basal diet was formulated according to the feed ingredient composition of the National Research Council (NRC, 2012), and its composition and nutritional levels are shown in Table 1. The experimental pig farm is on high terrain, is more than 800 m from country roads, is more than 1000 m from main roads, is open, and has no nearby pollution sources. The experimental room is a solid-wall, semi-permeable pig house with a 9-m span and a 30-m length arranged in two rows, with a middle walk of 1.05 m. In the room, each row has 17 delivery beds; the size of each bed is 2.2 m × 1.75 m. The pig house adopted a natural ventilation mode, the temperature was maintained at about 23 °C, and the humidity was maintained at about 80%. Manure and urine were manually cleaned in the pig house once a day, in the morning. During the experimental period, piglets were nursed normally, feed was provided in small amounts several times per day, and water was provided using automatic waterers.

### 2.3. Performance Indicators for Production

The weights of each experimental piglet, both at birth and seven days before weaning, were recorded. The feed weight (kg) and the lost feed weight (kg) of each group were recorded until seven days after weaning. The feed consumption and the average daily feed intake (ADFI, kg) were calculated. On the first day after the experiment ended, the piglets were weighed before feeding. The body weight gains and the average daily gains (ADG, kg) of the piglets were calculated.

### 2.4. Intestinal Sample Collection and Storage

On the eighth day after weaning, three piglets in each group were randomly selected for euthanasia (via the injection of sodium pentobarbital) and anatomical sampling. In each specimen, the abdominal cavity was opened, the small intestine was extracted, and the contents of the jejunum and ileum were removed and stored in a 2 mL sample tube. The samples were frozen for later use. In addition, two 2 cm intestinal segments, one from the jejunum and one from the ileum, were sampled; the surface residues were removed with tissue paper, the intestinal mucosae were scraped and stored in 2 mL sample tubes, and the samples were then frozen, at the temperature mentioned above, for the determination of intestinal morphology. Furthermore, from the middle parts of both the jejunum and the ileum of each specimen, two 2 cm segments were sampled and stored in a 5 mL sample bottle, and 5–10× volume 4% formaldehyde solution was added, fixed for later examination.

### 2.5. Measurement of Intestinal Indexes

Intestinal pH: The intestinal pH was measured with a TW-6226 pH meter (Beijing Tune Volt Water Instrument Technology Co., Ltd. Beijing, China).Enzyme Activity in the Intestinal Tract: The activities of α-amylase (U·mg^−1^), lipase (U·g^−1^), and trypsin (U·mg^−1^) in the intestines were determined by the corresponding kits. The kits were purchased from Wuhan Huamei Bioengineering Co., Ltd. (Wuhan China), and the operations were performed following the kit instructions.Intestinal Morphology using Hematoxylin and Eosin (H&E) Staining: The H&E staining procedure included the following steps: dehydration and clearing, wax embedding, sectioning and patching, dewaxing, dyeing, dehydration and clearing, and coverslipping. The H&E-stained tissue sections were observed under an optical microscope. Five good views for each section were selected, and Image-Pro Plus was used to measure and analyze the villus heights (mm), crypt depths (mm), villus heights/crypt depths (V/C), and mucosal thicknesses (mm). Where the villus height referred to the vertical distance from the villus tip to the crypt opening, the crypt depth was the vertical distance from the crypt opening to the crypt base, and the mucosal thickness was the vertical distance from the mucosal epithelium to the muscularis mucosae (including the muscularis).

### 2.6. Data Processing and Analysis

Microsoft Excel was used to record and organize the experimental data. SPSS 24.0 was adopted for statistical analysis. The results were expressed as “mean ± standard deviation” (Mean ± SD). Differences between groups were tested by one-way analysis of variance (ANOVA) followed by Fisher’s Least Significant Difference (LSD) test. A result of *p* > 0.05 was considered to be not statistically significant, and a result of *p* < 0.05 was considered statistically significant.

## 3. Results

### 3.1. Effects of Weaning Modes on Piglet Performance

The effects of weaning modes on piglet performance are shown in Table 2. As presented in the table, the weights at the end of the experiment, at an age of 36 days, had increased. The piglets in group M had the highest end weight, which was significantly higher than those of the other two groups (*p* < 0.05), whereas no significant difference was found between group C and group S (*p* > 0.05). Group M demonstrated significantly higher ADFI than that of groups C and S (*p* < 0.05), but there was no significant difference between groups C and S (*p* > 0.05). The effect of weaning modes on ADG was the same as it was on end weight. The ADG in group M was significantly higher than in the other two groups (*p* < 0.05), and no significant difference was found between group C and group S (*p* > 0.05).

### 3.2. Effects of Weaning Modes on Piglet Intestinal pH

The effects of different weaning modes on the intestinal pH of piglets are shown in Table 3. Based on the data presented in the table, the pH values in the jejuna of the three groups of piglets were not significantly different (*p* > 0.05). However, the ilea pH of group S was significantly lower than that of groups C and M (*p* < 0.05): 4.97% lower than group C (*p* < 0.05) and 2.39% lower than group M (*p* < 0.05). The ileum pH values of group M and group C were not significantly different (*p* > 0.05).

### 3.3. Effects of Weaning Modes on Digestive Enzyme Activities in Piglet Intestines

Table 4 shows the effects of weaning modes on digestive enzyme activities in piglet intestines. It can be observed that different weaning modes demonstrated an impact on the activities of three enzymes, namely, α-amylase, lipase, and trypsin, in the intestines of the piglets. The activities of the three enzymes in the two experimental groups were significantly higher than those in the control group C (*p* < 0.05). According to the activities of the three enzymes in the jejuna of the piglets, it can be observed that all were highest in group M, followed by group S, and finally group C (control) had the lowest. The activities of α-amylase, lipase, and trypsin in the jejuna of group M increased by 35.85%, 30.81%, and 19.00%, respectively, compared with group C (*p* < 0.05), and increased by 8.00%, 7.68%, and 7.00%, respectively, compared with group S (*p* < 0.05). The activities of α-amylase, lipase, and trypsin in the jejuna of group S increased by 25.79%, 21.49%, and 9.97%, respectively, compared with group C (*p* < 0.05). The activities of the three enzymes in the ilea were slightly different from those in the jejuna. However, the activity of α-amylase in the ilea of the three groups was the same as in the jejuna, with the highest activity in group M, followed by group S, then group C. The α-amylase activity in the ilea of group M piglets was elevated by 56.99% and 10.61% compared with groups C and S, respectively (*p* < 0.05). The α-amylase activity in the ilea of group S piglets was 41.94% higher than that of group C (*p* < 0.05), while the activities of lipase and trypsin in the ilea of group C piglets were the lowest. The activities of lipase and trypsin in the ilea of group M were elevated by 4.15% and 15.78%, respectively, compared with group C (*p* < 0.05), and those in group S were elevated by 28.00% and 13.05%, respectively, compared with group C (*p* < 0.05). No significant differences were shown in the activities of these two enzymes in the ilea of piglets in groups S and M (*p* > 0.05).

### 3.4. Effects of Weaning Modes on Piglet Intestinal Morphology

#### 3.4.1. Effects of Weaning Modes on Piglet Intestinal Villus Height, Crypt Depth, and Villus Height/Crypt Depth Ratio

The villi and crypts of the jejuna and ilea of piglets were different in morphology when influenced by different weaning modes. The villus height and crypt depth measurements are shown in Table 5. The jejunum villus height was the highest in group M, followed by group S, then group C, with significant differences between the three groups (*p* < 0.05). In detail, the jejuna villi heights in group M increased by 39.13% and 10.34% compared with those in groups C and S (*p* < 0.05), respectively, and those of group S increased by 26.09% compared with group C (*p* < 0.05). No significant difference in ilea villi heights was found between groups M and S (*p* > 0.05), but both were significantly higher than those of group C (*p* < 0.05). There was no significant difference in the crypt depths of the piglets among the three groups (*p* > 0.05). In terms of the ratio of villus height/crypt depth, no significant difference was found between groups M and S (*p* > 0.05), but both were significantly higher than those of group C (*p* < 0.05). In summary, the ratios of villus height/crypt depth in the jejuna and ilea of group M increased by 46.88% and 31.55%, respectively, compared with group C (*p* < 0.05). Those of group S increased by 41.41% and 19.25%, respectively, compared with group C (*p* < 0.05).

#### 3.4.2. Effects of Weaning Modes on the Thicknesses of Piglet Intestinal Mucosae

By measuring the mucosal thickness of the jejunum and ileum (Table 5), it was found that the jejunum mucosal thicknesses of groups M and S were significantly higher than those of group C (*p* < 0.05). In detail, the jejunum mucosal thicknesses of group M increased by 60.47% compared with those of group C (*p* < 0.05). Conversely, the thicknesses of group S increased by 30.23% compared with group C (*p* < 0.05), and no significant difference was found between groups M and S (*p* > 0.05). The highest ileum mucosal thicknesses were in group M, which were 15.38% higher than those of group C (*p* < 0.05) and 12.5% higher than those of group S (*p* < 0.05). There was no significant difference between groups C and S (*p* > 0.05).

## 4. Discussion

### 4.1. Effects of Weaning Modes on Piglet Performance

On one hand, a piglet’s production performance is affected by the sow’s milk production and the piglet’s hormone regulation. On the other hand, it is also impacted by environmental factors. Piglet mixing brings changes to the environment. Mixing modes under three different weaning modes had different effects on the performance of the piglets. The results revealed that both weaning after mixing and odorant spray modes significantly boosted piglet weight at the end of nursery (*p* < 0.05). This finding is consistent with the results of Zhang et al. [11].

### 4.2. Effects of Weaning Modes on Piglet Intestinal pH

pH is one of the essential factors affecting the digestive environments of animals. The value of pH directly affects the digestion and absorption of substances in the intestines. Studies have proven that the GI tract is not yet fully developed in piglets at the early weaning stage. At this stage, gastric acid is weak, lactic acid production in the intestines is low, and part of the gastric acid combines with feed ingredients, resulting in higher intestinal pH than at the preweaning level [12,13]. Studies have also demonstrated that lactic acid produced by the fermentation of lactose is an important substance for maintaining low acidity in the intestines, and intestinal pH is also an essential factor for supporting the production and activity of digestive enzymes [14]. Elevated pH sabotages the activity of digestive enzymes and disrupts the balance of GI microflora, leading to a disturbed environment, a reduced digestion and absorption area, a damaged intestinal mucosal barrier in the GI tract, and weakened immunity [15,16]. The results of this study demonstrated that, compared with the conventional weaning group, the intestinal pH of the odorant spray group and the weaning-after-mixing group both showed a decreasing trend, and the pH of the ilea in the odorant spray group was significantly different than those of the conventional weaning group (*p* < 0.05). This was consistent with the research results of Li (2017), who discovered that feeding piglets *Lactobacillus plantarum* during weaning improves intestinal development, lowers intestinal pH, promotes intestinal health, and relieves weaning stress [17]. The reasons for the above results may be because the weaning modes of odorant spraying and weaning after mixing can effectively counteract imbalances in GI microflora, internal environment disturbances, and decreased lactic acid production caused by stress.

### 4.3. Effects of Weaning Modes on Digestive Enzyme Activities in Piglet Intestines

The digestion of nutrients is closely related to the activities of digestive enzymes [18]. Most digestive enzymes are produced in the small intestine, and the activities of endogenous digestive enzymes directly affect the digestion and absorption capacity of the digestive system with respect to nutrition [19]. Studies have revealed that the activities of digestive enzymes in the gut are affected by various factors, including pH, nutrition, salinity, developmental stage, and diet [20]. Any factor that affects the physicochemical properties of the gut may have an impact on the activities of digestive enzymes [20]. The pH in a piglet’s gut is an essential factor in maintaining the production and activity of digestive enzymes, since many digestive enzymes have higher activities under acidic conditions [21,22]. At the age of 0–4 weeks, the activities of lipase and trypsin in a piglet’s gut double each week [23]. At one week after weaning, the activities of digestive enzymes decrease to one-third of those at the preweaning level, and it takes two weeks to recover to or exceed the preweaning level [23]. Therefore, the GI tracts of piglets have a low nutrient utilization rate regarding carbohydrates and fats in nondairy feed during the first week after weaning. At this stage, they cannot digest the proteins present in plant feed well or adapt to solid feeds. This causes indigestion and diarrhea in the first 1–2 weeks of early weaning [24]. The results of our study showed that, compared with the conventional weaning group, the activities of amylase, lipase, and trypsin in the jejuna and ilea of the odorant spray group and the weaning-after-mixing group were significantly elevated (*p* < 0.05). Our results are consistent with Wei et al. (2020), who found that adding resveratrol and curcumin to the diet can increase digestive enzyme activities in the small intestine and reduce weaning stress on weaned piglets [25]. The reasons for the above results may be that the weaning modes of odorant spraying and weaning after mixing can effectively countermeasure elevated pH, the disturbance of intestinal microflora, and the indigestion of nutrients caused by stress.

### 4.4. Effects of Weaning Modes on Piglet Intestinal Morphology

#### 4.4.1. Effects of Weaning Modes on Piglet Intestinal Villus Height, Crypt Depth, and Villus Height/Crypt Depth Ratio

Intestinal villus height, crypt depth, and villus height/crypt depth ratio are important indicators in determining gut health and integrity [26]. The higher the villi, the larger the absorption area of the small intestine and the higher the absorption efficiency. The greater the villus height/crypt depth, the stronger the intestinal absorption capacity [27]. Weaning stress can cause varying degrees of villus shortening, crypt deepening, and villus height/crypt depth decrease, resulting in reduced nutrient absorption and lowered growth performance [28,29]. The results of our study showed that, compared with the conventional weaning group, the intestinal villus heights and villus height/crypt depth ratios of the odorant spray group significantly increased (*p* < 0.05). These indicators were also significantly increased in the weaning-after-mixing group (*p* < 0.05), but the crypt depth became shallower. This was consistent with the results reported by Li et al. (2020), who found that adding an appropriate amount of porous zinc oxide to the diet increases the villus height of the jejunum and ileum and significantly increases the gut villus height/crypt ratio [30]. The above results may be due to the weaning modes of odorant spraying and weaning after mixing, effectively relieving stress-induced intestinal microflora composition changes, the indigestion of nutrients, and reduced intestinal antioxidant capacity.

#### 4.4.2. Effects of Weaning Modes on the Thickness of Piglet Intestinal Mucosae

Gut health is vital to the overall metabolism, physiology, disease defense, and growth and development of weaned piglets. Gut health includes the digestion and absorption of nutrients, the excretion of metabolic waste, the intestinal mucosal barrier, and the absence of GI illness. Intestinal villus height, crypt depth, and mucosal thickness represent the digestion and absorption capacity of the small intestine. In addition, mucosal thickness is related to the absorption and transportation of nutrients, which further affect the absorption function of the small intestine [31]. The results of this study demonstrated that, compared with the conventional weaning group, the jejunal mucosal thicknesses of the odorant spray group significantly increased (*p* < 0.05), and the jejunal and ileal mucosal thicknesses of the weaning-after-mixing group also significantly increased (*p* < 0.05). Nevertheless, our results were inconsistent with Yang’s (2016) study, which revealed that feeding piglets with various fermented soybean meals significantly reduces the mucosal thicknesses of the jejunum and ileum; however, this came with extremely significant increases in the mucosal thickness of the duodenum [32]. The underlying reason might be related to differences between individual piglets, differences in basal diets, and the influence of subjective factors during measurement.

## 5. Conclusions

Compared with conventional weaning modes, 2% (*v*/*v*) glutaraldehyde spray and weaning after mixing could increase the daily weight gain of piglets, lower the intestinal pH, boost digestive enzyme activity, and improve intestinal morphology. This indicates that these two weaning modes can reduce weaning stress to a certain extent, which is beneficial to the improvement of piglet performance. In addition, the mode of weaning after mixing performed optimally.

## Figures and Tables

**Table 1 animals-12-02200-t001:** Basal diet composition and nutritional levels (air-dried basis).

Feed	Content
Corn (g·kg^−1^)	605.0
Fish meal (g·kg^−1^)	50.0
Corn gluten meal (g·kg^−1^)	50.0
Soybean oil (g·kg^−1^)	10.0
Soybean meal (g·kg^−1^)	240.0
Limestone (g·kg^−1^)	11.8
CaHPO4 (g·kg^−1^)	13.0
L-Lys (g·kg^−1^)	6.0
Met (g·kg^−1^)	1.3
Thr (g·kg^−1^)	1.7
Ser (g·kg^−1^)	0.2
Choline chloride (g·kg^−1^)	1.0
NaCl (g·kg^−1^)	4.0
Premix (g·kg^−1^) ^1^	6.0
Analysis	
DE (MJ·kg^−1^)	14.1
CP (g·kg^−1^)	202.1
Ca (g·kg^−1^)	7.6
AP (g·kg^−1^)	4.5
Lys (g·kg^−1^)	12.5
Met (g·kg^−1^)	4.3
Thr (g·kg^−1^)	7.1
Ser (g·kg^−1^)	1.6

^1^ Premix provided per kg of diet: VA, 6000 IU; VD3, 400 IU; VE, 30 mg; VK3, 2 mg; VB1, 3.5 mg; VB2, 5.5 mg; VB6, 3.5 mg; VB12, 25.0 μg; biotin, 0.05 mg; folic acid, 0.3 mg; D-pantothenic acid, 20 mg; niacin, 20 mg; choline chloride, 500 mg; Fe, 110 mg; Zn, 100 mg; Cu, 20 mg; Mn, 40 mg; Se, 0.30 mg; and I, 0.40 mg. Nutrition levels were calculated values.

**Table 2 animals-12-02200-t002:** Effects of weaning modes on piglet performance.

Item	Group			*p*-Value
	Group C	Group S	Group M	
Original weight (kg)	5.67 ± 0.15 ^a^	5.60 ± 0.20 ^a^	5.63 ± 0.15 ^a^	0.523
Final weight (kg)	9.00 ± 0.43 ^a^	9.33 ± 0.16 ^a^	9.93 ± 0.32 ^b^	0.023
ADFI (kg)	0.61 ± 0.09 ^a^	0.63 ± 0.09 ^a^	0.66 ± 0.09 ^b^	0.014
ADG (kg)	0.24 ± 0.04 ^a^	0.27 ± 0.04 ^a^	0.31 ± 0.01 ^b^	0.013

Note: Different lowercase letters in the same row indicate significant differences between groups (*p* < 0.05), while no letters or the same letters indicate insignificant differences (*p* > 0.05). The same notation applies to Table 2, Table 3, Table 4 and Table 5.

**Table 3 animals-12-02200-t003:** Effects of weaning modes on piglet intestinal pH.

Item	Group			*p*-Value
	Group C	Group S	Group M	
Jejunum	6.32 ± 0.09 ^a^	6.12 ± 0.11 ^a^	6.24 ± 0.11 ^a^	0.569
Ileum	6.44 ± 0.19 ^b^	6.12 ± 0.18 ^a^	6.27 ± 0.02 ^b^	0.047

**Table 4 animals-12-02200-t004:** Effects of weaning modes on digestive enzyme activities in piglet intestines.

Item		Group			*p*-Value
		Group C	Group S	Group M	
α-Amylase (U·mg^−1^)	Jejunum	1.59 ± 0.01 ^a^	2.00 ± 0.06 ^b^	2.16 ± 0.02 ^c^	0.008
	Ileum	0.93 ± 0.04 ^a^	1.32 ± 0.02 ^b^	1.46 ± 0.02 ^c^	0.007
Lipase (U·g^−1^)	Jejunum	14.80 ± 0.34 ^a^	17.98 ± 0.51 ^b^	19.36 ± 0.35 ^c^	<0.001
	Ileum	52.54 ± 0.41 ^a^	67.25 ± 0.21 ^b^	70.04 ± 0.32 ^b^	0.006
Trypsin (U·mg^−1^)	Jejunum	800.46 ± 9.29 ^a^	890.24 ± 3.98 ^b^	952.55 ± 13.70 ^c^	<0.001
	Ileum	800.36 ± 8.25 ^a^	904.81 ± 10.97 ^b^	926.70 ± 6.00 ^b^	0.002

**Table 5 animals-12-02200-t005:** Effects of weaning modes on piglet intestinal morphology.

Item		Group			*p*-Value
		Group C	Group S	Group M	
Villus height (mm)	Jejunum	0.23 ± 0.01 ^a^	0.29 ± 0.02 ^b^	0.32 ± 0.01 ^c^	<0.001
	Ileum	0.30 ± 0.01 ^a^	0.38 ± 0.03 ^b^	0.37 ± 0.02 ^b^	<0.001
Crypt depth (mm)	Jejunum	0.18 ± 0.01 ^a^	0.16 ± 0.01 ^a^	0.17 ± 0.02 ^a^	0.121
	Ileum	0.16 ± 0.01 ^a^	0.17 ± 0.01 ^b^	0.15 ± 0.00 ^b^	<0.001
V/C	Jejunum	1.28 ± 0.07 ^a^	1.81 ± 0.08 ^b^	1.88 ± 0.19 ^b^	0.002
	Ileum	1.87 ± 0.12 ^a^	2.23 ± 0.07 ^b^	2.46 ± 0.11 ^b^	0.047
Mucosal (mm)	Jejunum	0.43 ± 0.01 ^a^	0.56 ± 0.04 ^b^	0.69 ± 0.05 ^b^	0.003
	Ileum	0.39 ± 0.01 ^a^	0.40 ± 0.01 ^a^	0.45 ± 0.04 ^b^	<0.001

## Data Availability

The data presented in this study are available upon request from the corresponding author.

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
