# Peer review of "Effects of Weaning Modes on the Intestinal pH, Activity of Digestive Enzymes, and Intestinal Morphology of Piglets"

_animals, 2022, doi:10.3390/ani12172200_

Round 1

Reviewer 1 Report

Dear editor 

The aim of the manuscript animals-1861716 entitled ‘’Effects of Weaning Modes on Intestinal pH, Activities of Digestive Enzymes, and Intestinal Morphology of Piglets’’ is relevant with the scope of the journal. 

The study is well conceived and written. The results of the study are useful for the pig farming and especially the effects of weaning modes on gut health of piglets.

Therefore, I suggest that the article be accepted for publication, under major revision.

Please let me congratulate you on the quality of your journal and thank you for giving me the opportunity to contribute as a reviewer.

Comments to the authors

Introduction 

·       L53-57: add appropriate references

·       L64-65: add appropriate references

Materials and Methods

§  L75-76: provide Ethical Note for your study, including the approval number by the Ethics Committee

§  L81: three replicates

§  L79-82: How many female and male piglets per group? were there any sex differences in the results? please add this information

§  L84: 7 consecutive days

§  L86: in situ (italics)

§  L86: 7 days

§  L98-100: rephrase the sentence

§  L107: body weight was determined in batches and not individually? please clarify. 

§  L107: body weight / 7 days

§  L109: 7 days

Discussion

§  L252: Li (2017)

§  L259-262: add appropriate references 

§  L278: Wei at al. (2020)

§  L315: Yang (2016)

Reviewer 2 Report

The purpose of this study was to investigate the effects of different weaning methods on alleviating weaning stress in piglets. The study found that two treatments of 2% (v/v) glutaraldehyde spray and weaning after mixing can effectively alleviate the weaning stress of piglets, which has certain guiding significance for animal production. Overall, this article is well-structured and logically clear. This manuscript is proposed for publication in Animals, with minor revisions. Specific comments/suggestions are as follows.

1. Line 21:Please highlight 'Table 1'.

2. Line 127-128:Note the correct writing of the unit symbol.

3. Line 149:Please highlight 'Table 2'.

4. Line 159-160:Please superscript the lowercase letters in Table 2 and make the same changes for Table 3-5.

5. Line 165: Please highlight 'Table 3'.

6. Line 173:Please highlight 'Table 4'.

7. Line 214-215: Please add '%'' after '46.88' and '41.41'.

8. Line 218: Please highlight 'Table 5'.

9. Line 228-234: Please change 'piglet' to 'piglets'.

10. Line 251-252:Note the tense of the sentence.

11. Line 262-264 : Please provide relevant references to support.

12. Line 267-268: Please provide relevant references to support.

13. Line 297-300: Note the tense of the sentence.